# Criminal networks analysis in missing data scenarios through graph distances

**Annamaria Ficara**[1,3☯], **Lucia Cavallaro**[2☯], **Francesco Curreri**[1,3☯], **Giacomo Fiumara**[3‡], **Pasquale De Meo**[4‡], **Ovidiu Bagdasar**[2‡], **Wei Song**[5‡], **Antonio Liotta**[6‡]*

**1** DMI Department, University of Palermo, Palermo, Italy, **2** School of Computing and Engineering, University of Derby, Derby, United Kingdom, **3** MIFT Department, University of Messina, Messina, Italy, **4** DICAM Department, University of Messina, Messina, Italy, **5** College of Information Technology, Shanghai Ocean University, Shanghai, China, **6** Faculty of Computer Science, Free University of Bozen-Bolzano, Bozen-Bolzano, Italy

☯ These authors contributed equally to this work.
‡ GF, PDM, OB, WS and AL also contributed equally to this work.
* Antonio.Liotta@unibz.it

**Data Availability Statement:** We have uploaded online on GitHub the source code of our algorithm. Link: https://github.com/lcucav/criminal-nets/tree/master/missing_data.

## Abstract

Data collected in criminal investigations may suffer from issues like: (i) incompleteness, due to the covert nature of criminal organizations; (ii) incorrectness, caused by either unintentional data collection errors or intentional deception by criminals; (iii) inconsistency, when the same information is collected into law enforcement databases multiple times, or in different formats. In this paper we analyze nine real criminal networks of different nature (i.e., Mafia networks, criminal street gangs and terrorist organizations) in order to quantify the impact of incomplete data, and to determine which network type is most affected by it. The networks are firstly pruned using two specific methods: (i) random edge removal, simulating the scenario in which the Law Enforcement Agencies fail to intercept some calls, or to spot sporadic meetings among suspects; (ii) node removal, modeling the situation in which some suspects cannot be intercepted or investigated. Finally we compute spectral distances (i.e., Adjacency, Laplacian and normalized Laplacian Spectral Distances) and matrix distances (i.e., Root Euclidean Distance) between the complete and pruned networks, which we compare using statistical analysis. Our investigation identifies two main features: first, the overall understanding of the criminal networks remains high even with incomplete data on criminal interactions (i.e., when 10% of edges are removed); second, removing even a small fraction of suspects not investigated (i.e., 2% of nodes are removed) may lead to significant misinterpretation of the overall network.

## Introduction

Criminal organizations can be defined as groups operating outside the boundaries of the law, that profit from providing illicit goods and services in public demand in an illicit manner, and for which achievements come at the detriment of other individuals, groups or societies [1]. Organized crime can be referred to by a range of different terms such as *gangs* [2], *crews* [3],

**Funding:** Our work is supported by the Open Access Publishing Fund of the Free University of Bozen-Bolzano.

**Competing interests:** The authors have declared that no competing interests exist.

*firms* [4], *syndicates* [4], or *Mafia* [5]. In particular, Mafia is defined in Gambetta's work [6] as a "territorially based criminal organization that attempts to govern territories and markets" and he refers to the one located in Sicily as the *original Mafia*.

Whatever term is used to identify organized crime, the latter is anyway based on relational traits. For this reason, scholars and practitioners are increasingly adopting a Social Network Analysis (SNA) perspective to explore criminal phenomena [7]. SNA is indeed a powerful tool to analyze criminal networks and to gain a deeper understanding of criminal behavior [8]. SNA algorithms are able to produce relevant measurements and parameters relevant to identify the roles and importance of individuals within criminal organizations [9] and to construct crime prevention systems [10].

Over the last decades, SNA has been employed greatly by Law Enforcement Agencies (LEAs). This increasing interest from law enforcement is due to the SNA ability to identify mechanisms that are not easily discovered at a first glance [11].

SNA relies on real datasets used as sources which allow to build networks that are then examined [10, 12–18]. However, the collection of complete network data describing the structure and activities of a criminal organization is difficult to obtain.

In a criminal investigation, the individuals subjected to LEAs enquiries may attempt to shield sensible information. Investigators then have to rely on alternative methods and exercise special investigative powers allowing them to gather evidence covertly. Information available for analysis can then come from sources such as phone taps, surveillance, archives, informants, interrogations to witnesses and suspects, infiltration in criminal groups. Despite significant advantages, such sources may also come with a number of drawbacks.

During investigations, some of the individuals providing information might be reliable, while others might attempt to deceive the investigations with the aim to protect themselves or their associates, or to achieve a goal. For instance, if actors are aware of being phone-tapped, they are more likely to avoid exposing some self-incriminating evidence. While transcripts of discussions between unsuspecting actors may be considered more reliable, a double-check is still needed between information collected from the taps and data collected from other official records related to the case. This is required since conversations among criminals often involve lies or codes concealing the true nature of the crime [19]. Moreover, if police misses surveillance targets, central actors may not appear with their actual role in the data, simply because their phones end up not being tapped [5].

While the police seeks to validate the content of phone-taps, the offenders themselves try to find out whether the information received from fellow criminals is actually accurate. Longer investigations and surveillance tend to eventually expose such lies. On the other side, with investigations going on, the list of suspects may change over time, with the group, and consequently data, changing significantly as a function of police decisions.

Police decisions may indeed impact the design of an investigation. LEAs normally start with some central individuals and then expand their reach by adding further actors. Not all the individuals linked to the central ones are automatically added, though. This can happen, for instance, when there are not enough resources available to investigate all active criminal groups, then prosecution services concentrate indeed on groups on which they can gather evidence easily. This kind of decisions are more prone to the risk of some groups operating under the police radar and then left out from the collected data. This approach is shown to hold extremely high chances of generating distorted inferences about the network structure [20].

The problem of actors lying is extended to data collected through questionnaires or interviews as well. Information collected from interrogations may not be reliable, with the risk of interviewees downplaying or amplifying their real role or not being representative of the broader group.

Incompleteness and incorrectness in criminal network data is then inevitable, since available intelligence data is determined more by the subjective judgements of investigators. This is due to investigators dealing with different qualities of data and because there is no standard methodology in SNA, for taking into account such degrees of reliability.

The problem of determining which information is relevant is usually referred to as the problem of *signal and noise*, in which important information is mixed in with large amounts of irrelevant, or unreliable information. LEAs are indeed often faced with the problem of having too much data, some of which being of little value. With large volumes of raw data collected from multiple sources, the risk of inconsistency becomes higher as well. Analytic techniques used in intelligence then must be able to cope with large amounts of information, and be capable to extract the signal from the noise.

In summary, data collected in criminal investigations often suffers from:

- *Incompleteness*, caused by the covert nature of such type of networks;

- *Incorrectness*, caused by either unintentional data collection errors and intentional deception by criminals;

- *Inconsistency*, when records of the same actors may be collected into law enforcement databases multiple times and not necessarily in a consistent way. Such misleading information may lead to an actor featuring multiple times (as different individuals) in the network.

Another problem specific to SNA for criminal networks lies in how data are transformed. As stated before, data needs to be presented in a specific manner, with actors being represented by nodes, whereas their associations or interactions are represented by links. In SNA, there is not a standard method for such data transformation task from raw data: the process undergoes the subjective judgement of the analyst that might be debatable. For instance, it may be difficult for an analyst to decide whom to include or exclude from the network, if its boundaries are prone to ambiguity [21]. Data conversion then ends up being a fairly labor-intensive and time-consuming task.

Finally, another feature of criminal networks is represented by their dynamics: such networks are not static, meaning that they constantly change over time. To represent such dynamics, new data or even different data collection methods are required, for covering longer time spans [21].

In this work, a network science approach is adopted to assess how much of the available data of a criminal network may be missing, before it starts to be unreliable. In other words, our aim is to quantify how much the partial knowledge of a criminal network can affect investigations in a significant way.

An interesting application of SNA consists of comparing networks, by finding and quantifying similarities and differences between them [22–24]. Network comparison requires measures for the distance between graphs, a non-trivial task involving sets of features which are often sensitive to the specific application domain. Some reviews on the most common graph comparison metrics are [25–28]. In [29], such distance measures were exploited to quantify how well artificial (but realistic) models can simulate real criminal networks. The same measures are used herein for a different task.

In this paper, we analyze nine real criminal networks of different nature, which are the result of different investigative operations over Mafia networks, criminal street gangs and terrorist organizations. To quantify the impact of incomplete data and to determine what kind of network mostly suffers from it, we adopted the following strategy:

1. We pruned input networks by means of two specific methods, namely: *random edge removal* and *random node removal*, which reflect the most common scenarios of missing data arising in investigation environments.

2. We calculated the distance between the original (and complete) network and its pruned version.

## Materials and methods

This section presents basic definitions and notations on graph theory concepts and the distance metrics used for comparing two graphs. We also describe the datasets used in our experimental analysis, as well as the protocol followed to run our analysis.

### Background

**Graph properties.**   A *network* (or *graph*) $G = \langle N, E \rangle$ consists of two finite sets $N$ and $E$ [30]. The set $N = \{1, \ldots, n\}$ contains the *nodes* (or vertices, actors), and $n$ is the *size* of the network, while the set $E \subseteq N \times N$ contains the *edges* (or links, ties) between the nodes.

A network is called *undirected* if all its edges are bidirectional. If the edges are defined by ordered pairs of nodes, then the network is called *directed*. If an edge $(i, j)$ with $i, j \in N$ is *weighted*, then a positive numerical weight $w_{ij}$ is associated to it; the *unweighted* edges have their weight set to the default value $w_{ij} = 1$.

Given an undirected network $G$, two nodes $i, j \in N$ are *connected* if there is a *path* from $i$ to $j$: here a path $p$ is defined as a sequence of nodes $i_0, i_1, \ldots, i_k$ such that each pair of consecutive nodes is connected through an edge. The number of edges in a path $p$ starting at node $i$ and ending at node $j$ is called *path length*. While there may be several paths from the node $i$ to the node $j$, we are usually interested in the *shortest paths* (i.e., those with the least number of edges), whose length defines the *distance* $d_{ij}$ between $i$ and $j$. Of course, in undirected networks we have $d_{ij} = d_{ji}$.

A graph $G$ is called *connected* if every pair of nodes in $G$ is connected, and *disconnected* otherwise. If a network is disconnected, then this can be partitioned into a collection of connected subnetworks, called *components*.

Based on the number of edges $m$, a graph is called *dense* if $m$ is of the same order of magnitude as $n^2$, or *sparse* if $m$ is of the same order of magnitude as $n$. The *density* $\delta$ of an undirected graph is defined as

$$\delta = \frac{2|E|}{|V|(|V| - 1)} = \frac{2m}{n(n - 1)}, \tag{1}$$

that is the total number of edges over the maximum possible number of edges.

The degree $k_i$ of the node $i$ represents the number of adjacent edges to it. For an integer $0 \leq k \leq n$, $n_k$ is the number nodes of degree $k$, while the degree distribution $p_k$ is the probability that a randomly selected node in the graph has degree $k$. For a graph of $n$ nodes, the normalized histogram for $p_k$ is then given by

$$p_k = \frac{n_k}{n}. \tag{2}$$

The degree $k_i$ allows to compute the *clustering coefficient $C_i$* of a node $i$ [31], which captures the degree to which the neighbors of the node $i$ link to each other, given by

$$C_i = \frac{2L_i}{k_i(k_i - 1)}, \tag{3}$$

where $L_i$ represents the number of links between the $k_i$ neighbors of node $i$. The average of $C_i$ over all nodes defined the average clustering coefficient $\langle C_i \rangle$, measuring the probability that two neighbors of a randomly selected node link to each other.

Given a pair of graphs, say $G_1$ and $G_2$, we are often interested in defining a measure of similarity (or, equivalently, distance) between them. In what follows we review some methods one can use to compute the distance of two graphs.

**Spectral distances.** Spectral distances allow to measure the structural similarity between two graphs starting from their spectra. The spectrum of a graph is widely used to characterise its properties and to extract information from its structure.

The most common matrix representations of a graph are the *adjacency matrix A*, the *Laplacian matrix L*, and the *normalized Laplacian $\mathcal{L}$*.

Given a graph $G$ with $n$ nodes, its adjacency matrix $A$ is an $n \times n$ square matrix denoted by $A = (a_{ij})$, with $1 \leq i, j \leq n$, where $a_{ij} = 1$ if there exists an edge joining nodes $i$ and $j$, and $a_{ij} = 0$ otherwise.

For undirected graphs the adjacency matrix is symmetric, i.e., $a_{ij} = a_{ji}$.

The degree matrix $D$ is a diagonal matrix where $D_{ii} = k_i$ and $D_{ij} = 0$ for $i \neq j$.

$$D_{ij} = \begin{cases} k_i & \text{if } i = j \\ 0 & \text{otherwise.} \end{cases} \tag{4}$$

The adjacency matrix and the degree matrix are used to compute the combinatorial Laplacian matrix $L$, which is an $n \times n$ symmetric matrix defined as

$$L = D - A. \tag{5}$$

The diagonal elements $L_{ii}$ of matrix $L$ are then equal to the degree $k_i$ of the node $i$, while the off-diagonal elements $L_{ij}$ are $-1$ if the node $i$ is adjacent to $j$, and $0$ otherwise. A normalized version of the Laplacian matrix $\mathcal{L}$ is defined as

$$\mathcal{L} = D^{-\frac{1}{2}} L D^{-\frac{1}{2}}, \tag{6}$$

where the diagonal matrix $D^{-\frac{1}{2}}$ is given by

$$D_{i,i}^{-\frac{1}{2}} = \begin{cases} \frac{1}{\sqrt{k_i}} & \text{if } k_i \neq 0 \\ 0 & \text{otherwise.} \end{cases} \tag{7}$$

If the representation matrix is symmetric, its eigenvalues are real and they can be sorted. The spectrum of a graph consists indeed of the set of the sorted eigenvalues of one of its representation matrices. The sequence of eigenvalues may be ascending or descending depending on the chosen matrix. The spectra derived from each representation matrix may reveal different properties of the graph. The largest eigenvalue in modulus is called the *spectral radius* of the graph. If $\lambda_k^A$ is the $k^{th}$ eigenvalue of the adjacency matrix $A$, then the spectrum is given by

the descending sequence

$$\lambda_1^A \geq \lambda_2^A \geq \cdots \geq \lambda_n^A. \tag{8}$$

If $\lambda_k^L$ is the $k^{th}$ eigenvalue of the Laplacian matrix $L$, such eigenvalues are considered in ascending order so that

$$0 = \lambda_1^L \leq \lambda_2^L \leq \cdots \leq \lambda_n^L. \tag{9}$$

The *second smallest eigenvalue* of the Laplacian matrix of a graph is called its *algebraic connectivity*. Similarly, if we denote the $k^{th}$ eigenvalue of the normalized Laplacian matrix $\mathcal{L}$ as $\lambda_k^{\mathcal{L}}$, then its spectrum is given by

$$0 = \lambda_1^{\mathcal{L}} \leq \lambda_2^{\mathcal{L}} \leq \cdots \leq \lambda_n^{\mathcal{L}}. \tag{10}$$

The *spectral distance* between two graphs is the euclidean distance between their spectra [32]. Given two graphs $G$ and $G'$ of size $n$, with their spectra respectively given by the set of eigenvalues $\lambda_i$ and $\lambda_i'$, their spectral distance, according to the chosen representation matrix, is computed as follows by the formula

$$d(G, G') = \sqrt{\sum_{i=1}^{n}(\lambda_i - \lambda_i')^2}. \tag{11}$$

Based on the chosen representation matrix and consequently its spectrum, the most common spectral distances are the adjacency spectral distance $d_A$, the Laplacian spectral distance $d_L$ and the normalized Laplacian spectral distance $d_{\mathcal{L}}$.

If the two spectra are of different sizes, the smaller graph is brought to the same cardinality of the other by adding zero values to its spectrum. In such case, only the first $k \ll n$ eigenvalues are compared. Given the definitions of spectra of the different matrices, the adjacency spectral distance $d_A$ compares the largest $k$ eigenvalues, while $d_L$ and $d_{\mathcal{L}}$ compare the smallest $k$ eigenvalues. This determines the scale at which the graphs are studied, since comparing the higher eigenvalues allows to focus more on global features, while the other two allow to focus more on local features.

**Matrix distances.** Another class of distances between graphs is the matrix distance [33]. A matrix of pairwise distances $d_{ij}$ between nodes on the single graph is constructed for each as

$$M_{ij} = d_{ij}. \tag{12}$$

While the most common distance $d$ is the shortest path, other measures can also be used, such as the effective graph resistance, or variations on random-walk distances. Such matrices provide a signature of the graph characteristics and carry important structural information. Matrices $M$ are then compared using some norm or distance.

Given two graphs $G$ and $G'$, having $M$ and $M'$ as their respective matrices of pairwise distances, the matrix distance between the $G$ and $G'$ is introduced as:

$$d(G, G') = \| M - M' \|, \tag{13}$$

where $\|.\|$ is a norm to be chosen. If the matrix used is the adjacency matrix $A$, the resulting distance is called *edit distance*.

The similarity measure used in this work is called DELTACON [34]. It is based on the root euclidean distance $d_{\text{rootED}}$, also called *Matsusita difference*, between matrices $S$ created from the fast belief propagation method of measuring node affinities.

The DELTACON similarity $sim_{DC}$ is defined as

$$sim_{DC}(G, G') = \frac{1}{1 + d_{rootED}(G, G')}, \qquad (14)$$

where the root euclidean distance $d_{rootED}(G, G')$ is defined as

$$d_{rootED}(G, G') = \sqrt{\sum_{i,j}\left(\sqrt{S_{i,j}} - \sqrt{S'_{i,j}}\right)^2}. \qquad (15)$$

When used instead of the Euclidean distance, $d_{rootED}(G, G')$ may even detect small changes in the graphs. The fast belief propagation matrix $S$ is defined as

$$S = [I + \varepsilon^2 D - \varepsilon A]^{-1}, \qquad (16)$$

where $\varepsilon = 1/(1 + \max_{1 \leq i \leq n} D_{ii})$ and it is assumed to be $\varepsilon \ll 1$, so that S can be rewritten in a matrix power series as:

$$S \approx I + \varepsilon A + \varepsilon^2(A^2 - D) + \dots. \qquad (17)$$

Fast belief propagation is an effective algorithm and it is designed to perceive both global and local structures of the graph [34].

## Criminal networks data sources

Our analysis focuses on nine real criminal networks of different nature (see Table 1). The first six networks relate to three distinct Mafia operations, while the other three are linked to street gangs and terrorist organizations.

The Montagna Operation was an investigation concluded in 2007 by the Public Prosecutor's Office of Messina (Sicily) focused on the Sicilian Mafia groups known as Mistretta and Batanesi clans. Between 2003 and 2007 these families infiltrated several economic activities including public works in the area, through a cartel of entrepreneurs close to the Sicilian

**Table 1. Criminal networks characterization.**

| Investigation | Network | | | Source |
|---|---|---|---|---|
| | **Name** | **Nodes** | **Edges** | |
| Montagna Operation (Sicilian Mafia) 2003-2007 | MN PC | Suspects | Physical Surveillance Audio Surveillance | [10, 16, 17, 29, 35] |
| Infinito Operation (Lombardian 'Ndrangheta) 2007-2009 | SN | Suspects | Physical and Audio Surveillance | [36–40] |
| Oversize Operation (Calabrian 'Ndrangheta) 2000-2009 | WR AW JU | Suspects | Audio Surveillance Physical Surveillance Audio Surveillance | [41, 42] |
| Swedish Police Operation (Stockholm Street Gangs) 2000-2009 | SV | Gang members | Physical Surveillance | [13, 43] |
| Caviar Project (Montreal Drug Traffickers) 1994-1996 | CV | Criminals | Audio Surveillance | [5] |
| Abu Sayyaf Group (Philippines Kidnappers) 1991-2011 | PK | Kidnappers | Attacks locations | [44] |

**Table 2. Mafia networks properties.**

| Network | MN | PC | SN | WR | AW | JU |
|---|---|---|---|---|---|---|
| weights | weighted | weighted | weighted | unweighted | unweighted | unweighted |
| directionality | undirected | undirected | undirected | undirected | undirected | undirected |
| connectedness | false | false | false | false | false | false |
| no. of nodes $n$ | 101 | 100 | 156 | 182 | 182 | 182 |
| no. of isolated nodes $n_i$ | 0 | 0 | 5 | 0 | 36 | 93 |
| no. of edges $m$ | 256 | 124 | 1619 | 247 | 189 | 113 |
| no. of components $|cc|$ | 5 | 5 | 6 | 3 | 38 | 96 |
| max avg. path length $\langle d \rangle$ for $cc$ | 3.309 | 3.378 | 2.361 | 3.999 | 4.426 | 3.722 |
| max shortest path length $d$ | 7 | 7 | 5 | 8 | 9 | 7 |
| density $\delta$ | 0.051 | 0.025 | 0.134 | 0.015 | 0.011 | 0.007 |
| avg. degree $\langle k \rangle$ | 5.07 | 2.48 | 20.76 | 2.71 | 2.08 | 1.24 |
| max degree $k$ | 24 | 25 | 75 | 32 | 29 | 13 |
| avg. clust. coeff. $\langle C \rangle$ | 0.656 | 0.105 | 0.795 | 0.149 | 0.122 | 0.059 |

Mafia. The main data source is the pre-trial detention order issued by the Preliminary Investigation Judge of Messina on March 14, 2007.

The order concerned a total of 52 suspects, all charged with the crime of participation in a Mafia clan as well as other crimes such as theft, extortion or damaging followed by arson. From the analysis of this legal document we built two weighted and undirected graphs: the Meeting network (MN) with 101 nodes and 256 edges, and the Phone Calls (PC) network with 100 nodes and 124 edges (see Table 2). In both networks, nodes are suspected criminals and edges represent meetings (MN), or recorded phone calls (PC). These original datasets have been already studied in some of our previous works [10, 16–18, 29, 45] and they are available on Zenodo [35].

The Infinito Operation was a large law enforcement operation against 'Ndrangheta groups (i.e., groups of the Calabrian Mafia) and Milan cosche (i.e., crime families or clans) concluded by the courts of Milan and Reggio Calabria, Italian cities situated in Northern and Southern Italy, respectively. The investigation started 2003 is still in progress. On July 5, 2010, the Preliminary Investigations Judge of Milan issued a pre-trial detention order for 154 people, with charges ranging from mafia-style association to arms trafficking, extortion and intimidation for the awarding of contracts or electoral preferences. The dataset was extracted from this judicial act and is available as a 2-mode matrix on the UCINET [46] website (Link: https://sites.google.com/site/ucinetsoftware/datasets/covert-networks/ndranghetamafia2). The Infinito Operation dataset was investigated by Calderoni and his co-authors in several works [36–40]. From the original 2-mode matrix, we constructed the weighted and undirected graph Summits Network (SN) with 156 nodes and 1619 edges (Table 2). Nodes are suspected members of the 'Ndrangheta criminal organization. Edges are summits (i.e., meetings whose purpose is to make important decisions and/or affiliations, but also to solve internal problems and to establish roles and powers) taking place between 2007 and 2009. This network describes how many summits any two suspects may have in common. Attendance at summits was registered by police authorities through wiretapping and observations during this operation.

The Oversize Operation is an investigation lasting from 2000 to 2006, which targeted more than 50 suspects of the Calabrian 'Ndrangheta involved in international drug trafficking, homicides, and robberies. The trial led to the conviction of the main suspects from 5 to 22 years of imprisonment between 2007-2009. Berlusconi et al. [41] studied three unweighted and undirected networks extracted from three judicial documents corresponding to three

**Table 3. Street gangs and terrorist networks properties.**

| Network | SV | CV | PK |
|---|---|---|---|
| weights | weighted | weighted | weighted |
| directionality | undirected | undirected | undirected |
| connectedness | false | true | false |
| no. of nodes $n$ | 234 | 110 | 246 |
| no. of isolated nodes $n_i$ | 12 | 0 | 16 |
| no. of edges $m$ | 315 | 205 | 2571 |
| no. of components $|cc|$ | 13 | 1 | 26 |
| max avg. path length $\langle d \rangle$ for $cc$ | 3.534 | 2.655 | 3.034 |
| max shortest path length $d$ | 6 | 5 | 9 |
| density $\delta$ | 0.012 | 0.034 | 0.085 |
| avg. degree $\langle k \rangle$ | 2.69 | 3.73 | 20.9 |
| max degree $k$ | 34 | 60 | 78 |
| avg. clustering coeff. $\langle C \rangle$ | 0.15 | 0.335 | 0.753 |

different stages of the criminal proceedings (Table 2): wiretap records (WR), arrest warrant (AW), and judgment (JU). Each of these networks has 182 nodes corresponding to the individuals involved in illicit activities. The WR network has 247 edges which represent the wiretap conversations transcribed by the police and considered relevant at first glance. The AW network contains 189 edges which are meetings emerging from the physical surveillance. The JU network has 113 edges which are wiretap conversations emerging from the trial and several other sources of evidence, including wiretapping and audio surveillance. These datasets are available as three 1-mode matrices on Figshare [42].

The Stockholm street gangs dataset was extracted from the National Swedish Police Intelligence (NSPI), which collects and registers the information from different kinds of intelligence sources to identify gang membership in Sweden. The organization investigated here is a Stockholm-based street gang localised in southern parts of Stockholm County, consisting of marginalised suburbs of the capital. All gang members are male with high levels of violence, thefts, robbery and drug-related crimes. Rostami and Mondani [13] constructed the Surveillance (SV) network (Table 3). It contains data from the General Surveillance Register (GSR) which covers the period 1995–2010 and aims to facilitate access to the personal information revealed in law enforcement activities needed in police operations. SV is a weighted network with 234 nodes that are gang members. Some of them were no longer part of the gang in the period covered by the data and have been included as isolated nodes. The link weight counts the number of occurrence of a given edge. This dataset is available on Figshare [43].

Project Caviar [5] was a unique investigation against hashish and cocaine importers operating out of Montreal, Canada. The network was targeted between 1994 and 1996 by a tandem investigation uniting the Montreal Police, the Royal Canadian Mounted Police, and other national and regional law-enforcement agencies from England, Spain, Italy, Brazil, Paraguay, and Colombia. In a 2-year period, 11 imported drug consignments were seized at different moments and arrests only took place at the end of the investigation. The principal data sources are the transcripts of electronically intercepted telephone conversations between suspects submitted as evidence during the trials of 22 individuals. Initially, 318 individuals were extracted because of their appearance in the surveillance data. From this pool, 208 individuals were not implicated in the trafficking operations. Most were simply named during the many transcripts of conversations, but never detected. Others who were detected had no clear participatory role within the network (e.g., family members or legitimate entrepreneurs). The final Caviar (CV)

network was composed of 110 nodes. The 1-mode matrix with weighted and directed edges is available on the UCINET [46] website. (Link: https://sites.google.com/site/ucinetsoftware/datasets/covert-networks/caviar). From this matrix, we extracted an undirected and weighted network with 110 nodes which are criminals and 205 edges which represent the communications exchanges between them (see Table 3). Weights are level of communication activity.

Philippines Kidnappers data refer to the Abu Sayyaf Group (ASG) [44], a violent non-state actor operating in the Southern Philippines. In particular, this dataset is related to the Salast movement that has been founded by Aburajak Janjalani, a native terrorist of the Southern Philippines in 1991. ASG is active in kidnapping and other kinds of terrorist attacks. The reconstructed 2-mode matrix is available on UCINET [46] (Link: https://sites.google.com/site/ucinetsoftware/datasets/covert-networks/philippinekidnappings). From the 2-mode matrix, we constructed a weighted and undirected graph called Philippines Kidnappers (PK) (see Table 3). The PK network has 246 nodes and 2571 edges. Nodes are terrorist kidnappers of the ASG. Edges are the terrorist events they have attended. This network describes how many events any two kidnappers have in common.

Useful information about Mafia, street gangs and terrorist networks is provided in Tables 2 and 3, including edges weight and directionality, connectedness, number of nodes including isolated ones, number of edges, number of connected components, maximum average path length for each connected component, maximum shortest path length, average degree, maximum degree and the average clustering coefficient. The CV network seems to be the only fully connected network (i.e., $|cc| = 1$) and, for this reason, in all the considered networks we chose to compute the average path length for the single components and then to show the maximum value.

Then, we showed the degree distributions for each criminal network as a normalized histogram (see Fig 1). MN, PC, WR, AW, JU, SV and CV have similar degree distributions in which most nodes have a relatively small degree $k$ with values around 0, 1 or 2, while a few nodes have very large degree $k$ and are connected to many other nodes. SN and PK are the only networks having different degree distributions compared to other criminal networks, as most of their nodes have large degree $k$. In particular, we note that most nodes in PK are strongly connected and have a degree $k = 57$.

SN, which derives from the Infinito operation, is a one-mode projection of the original two-mode network in which are represented the meetings and the suspects attending them. This implies that all suspects taking part in a meeting are assumed to be interacting with each other, which could be somewhat artificial. In fact, in crowded meetings some participants may have had a very limited (if any) interaction with other participants. In such case, assuming that all participants interacted with each other may considerably overestimate the real number of connections. However, it must be added that LEAs were only able to identify the participants to meetings and not the full extent of their interactions. Similar consideration applies to PK which was built based on the presence of the kidnappers in the same place of a terrorist event. Here as well, the existence of an edge linking two terrorists does not necessarily imply that they have interacted or worked together, despite being in the same place.

## Design of experiments

In this section we give technical details on the design of the experiments conducted.

In the attempt of gaining a deeper understanding of criminal networks, in our previous work [29] we used graph distances to compare randomly generated graphs and a real criminal network. In the present paper, we have implemented distances, to understand the extent by which a partial knowledge of a criminal network may negatively affect the investigations. Since

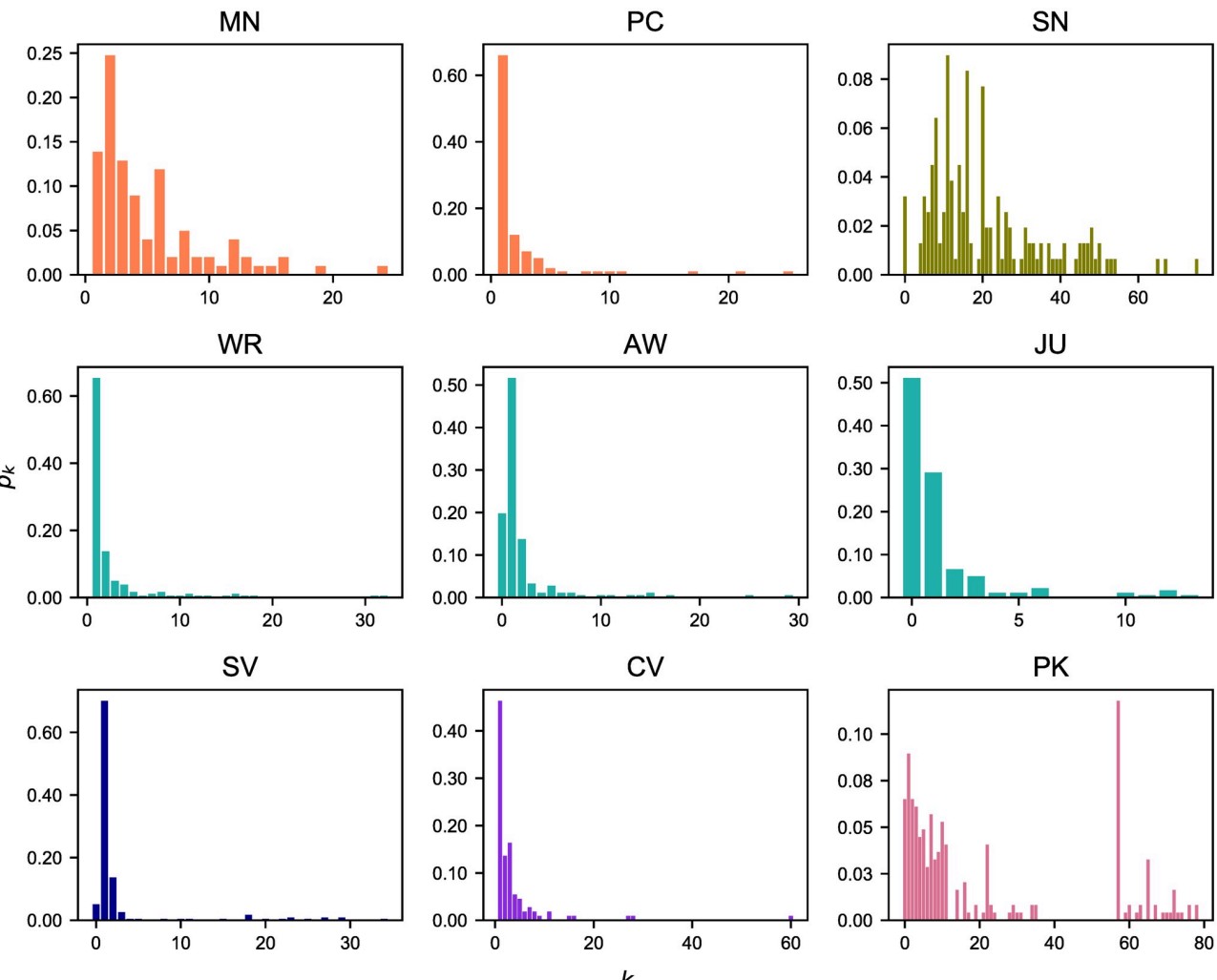

**Fig 1. Degree distributions.** The degree distribution $p_k$ provides the probability that a randomly selected node in each criminal network has degree $k$. Same colors imply the networks belong to the same police investigation.

we are trying to estimate differences based on the types/amount of data missing, we set up the experiments based on two main strategies: random edge removal and node removal. The first case simulates the scenario in which LEAs miss to intercept some calls or to spot sporadic meetings among suspects (i.e., due to the delays in obtaining a warrant). In node removal, the selected nodes are removed along with their incident edges, and afterwards they are reinserted within the networks as isolated nodes. Indeed, the second case models the scenario in which some suspects cannot be intercepted. For instance, if a criminal is known to be a boss but there are not enough proofs to be investigated, then that criminal can be identified as an isolated node with no incident edges. However, node removal is expected to have a greater impact than simple edge removal, since removing a node implies the deletion of all its edges as well.

Note that for a better comparison among the networks, all the graphs have been considered as unweighted (as AW and JU). Also, all the suspects showed as isolated nodes of the original network have been excluded. In fact, our input parameter was the edge list of the graph, which does not take into account nodes with no incident edges.

Algorithm 1 shows the pseudocode of our approach. The full code is available at https://github.com/lcucav/criminal-nets/tree/master/missing_data. In order to obtain the subgraphs, we started from the previously described datasets; then, we converted them into graphs (i.e., $G$) and, lastly, we pruned them (i.e., $G'$) according to a prefixed range of fractions with $0 < torem \leq 10\%$. We opted for the 10% because the criminal networks considered are small, as they have less than 250 nodes. Afterwards, we have computed the spectral and matrix distances between the original and the pruned graphs. Each edge removal process has been repeated a fixed number of times ($nrep$ = 100) and the results obtained have been averaged. Thus, the averaged distances values $\langle X \rangle$ and their standard deviations $\sigma$ have been computed.

**Algorithm 1** Pseudocode for computing the distances

```
 1: Parameter configuration: nrep, torem, and check
 2: Read the dataset and convert it as graph G
 3: if check = True then
 4:   Isolate torem of nodes
 5: else
 6:   Remove torem of random edges
 7: end if
 8: Compute S(G)
 9: Compute the matrices A(G), L(G), 𝓛(G)
10: for torem do
11:   for nrep do
12:     Create a pruned graph G′ and compute S′ (G′)
13:     Compute d_rootED(G, G′), d_A(G, G′), d_L(G, G′), and d_𝓛(G,G′)
14:   end for
15:   Compute ⟨X⟩, σ ∀ d(G, G′) ∈ nrep
16: end for
```

## Results

Here we present the results obtained from the network pruning experiments. The distance analysis between the real and the pruned networks is reported starting from the random edge removal approach (Fig 2), moving to the analysis on the networks after node pruning (Fig 3). The plots show the distances between the original graphs and their pruned versions up to 10% of edges ($F_e$) and nodes ($F_n$), respectively.

In both removal processes, $d_A$ displays a saturation effect that makes the results difficult to be interpreted. Hence, this distance is not effective for highlighting the effects of missing data on criminal networks. Furthermore, from this metric it might seem that the two pruned networks of PK and SN show a greater deviation from their original counterparts, but this is due to the inner structure of this metric, which is highly influenced by the node degree. In fact, the average degree of PK an SN (see Tables 2 and 3) is significantly higher (i.e., $\langle k \rangle \simeq 21$) than the other networks herein studied (i.e., $1 < \langle k \rangle < 4$); moreover, their different topology is also evident from their degree distribution (see Fig 1). This is the reason why these networks seem to have a more significant detachment effect than others; however, they too suffer the saturation effect mentioned above as they grow. A similar behavior has also been encountered in $d_L$ and its explanation is the same.

On the other hand, the distance metric which more effectively catches the damage caused by a significant amount of missing data is $d_{\mathcal{L}}$, where distance growth is linear. Indeed, the effects of $\langle k \rangle$ are smaller as this aspect is compressed by the structure of this distance metric. It would seem that this metric is the most effective measure compared to other spectral distances, in understanding how much lacking data affects the total knowledge of the network. A similar trend was also found in $d_{rootED}$; however, for a better comparison between node and edge

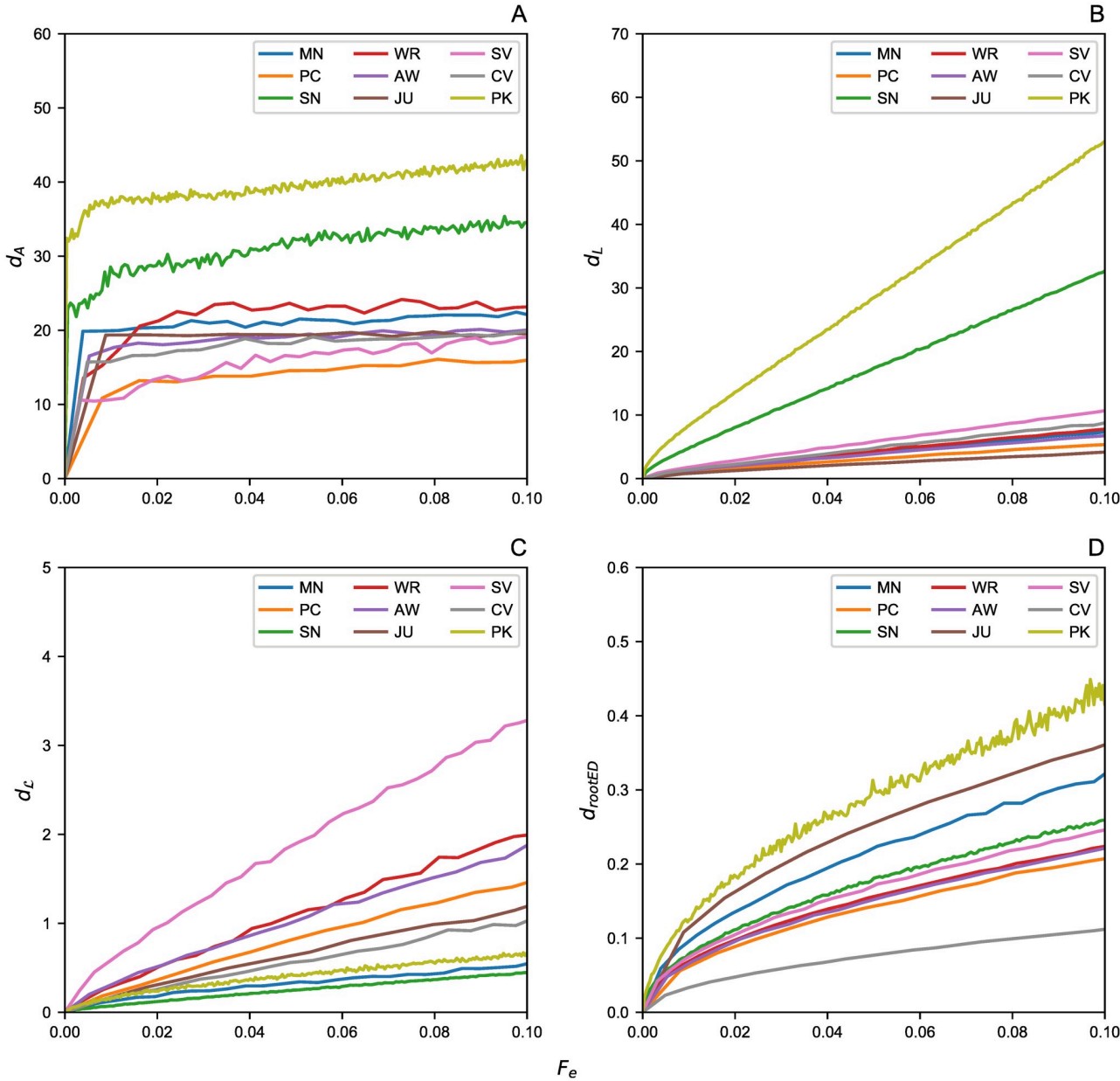

**Fig 2. Edge removal effect.** The removal effects of a fraction $F_e$ of edges by showing the graph distances between the original graphs with their pruned versions. (A) Adjacency Spectral Distance $d_A$. (B) Laplacian Spectral Distance $d_L$. (C) Normalized Laplacian Spectral Distance $d_{\mathcal{L}}$. (D) Root Euclidean Distance $d_{rootED}$.

removal processes, we analyzed in more detail this last metric by considering the DELTACON similarity $sim_{DC}$ (Fig 4).

The figure shows the difference between the original and pruned networks as the fraction of elements removed increases (i.e., $F_e$ for edges and $F_n$ for nodes).

Before pruning the networks we have $sim_{DC} = 1$. Afterwards, the drop begins to became more evident as the fraction $F$ increases. In addition, as expected, the node removal process affects more significantly the networks. This means that if the lack of data relates to sporadically missed wiretaps, or to just a few random connections between suspects, then the network

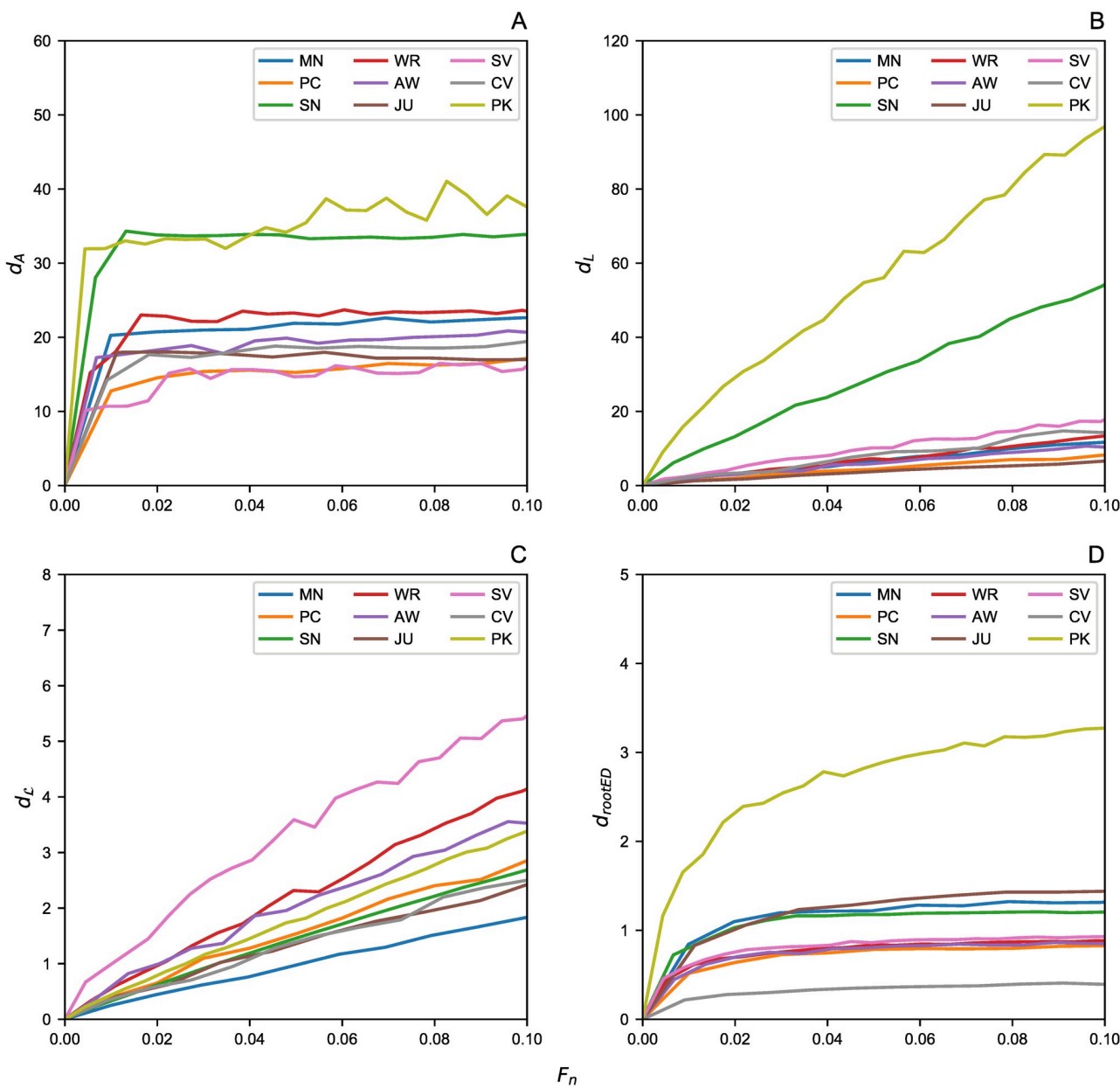

**Fig 3. Node removal effect.** The removal effects of a fraction $F_n$ of nodes by showing the graph distances between the original graphs with their pruned versions. (A) Adjacency Spectral Distance $d_A$. (B) Laplacian Spectral Distance $d_L$. (C) Normalized Laplacian Spectral Distance $d_\mathcal{L}$. (D) Root Euclidean Distance $d_{rootED}$.

structure is not as much misinterpreted as if the case when one suspect has not been tracked at all. Indeed, pruning the network by 2%, causes a $sim_{DC} \geq 0.8$ for edge pruning, compared to a $sim_{DC} \simeq 0.2$ for the nodes ones. Therefore, even when a small amount of suspects are not included in the investigations, this can lead to a very different network. The exclusion of the suspects could be voluntary or not. It highly depends on the overall investigation process, starting from the very preliminary analysis, and up to the judges' decision to allow warrants, or to exclude data considered irrelevant for the current investigation.

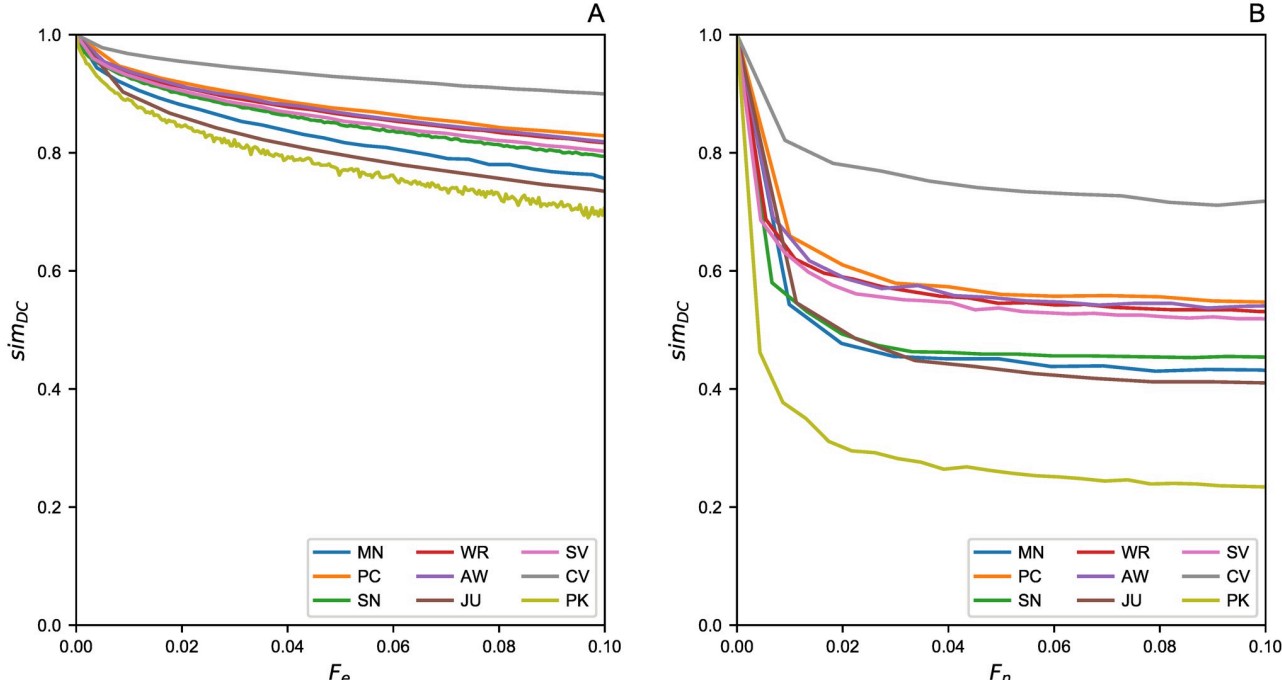

**Fig 4. DeltaCon similarity $sim_{DC}$ computation.** (A) Edge removal process by the fraction $F_e$. (B) Node removal process by the fraction $F_n$.

## Discussion

In this paper we analyzed nine datasets of real criminal networks extracted from six police operations to investigate the effects of missing data. More specifically, three datasets regard Mafia operations (i.e., Montagna, Infinito, and Oversize), and the remaining ones refer to other criminal networks, including street gangs, drug traffics, or terrorist networks (i.e., Stockholm street gangs, Caviar Project, Philippines Kidnappers).

Our study focused on a careful analysis of the datasets, in order to simulate the events where some data are missing. In particular, two different scenarios have been considered: (i) random edge removal, simulating the case in which LEAs miss to intercept some calls or to spot sporadic meetings among suspects; and (ii) node removal, for the scenario where certain suspects cannot be intercepted for some reason. For instance, if a criminal is known to be a boss, but there are not enough proofs for him or her to be investigated, then this can be identified by an isolated node with no incident edges.

To quantify the difference between the original criminal networks and their pruned counterparts, several distance metrics have been considered. We computed the Adjacency, Laplacian, and normalized Laplacian Spectral distances (i.e., $d_A$, $d_L$, and $d_{\mathcal{L}}$, respectively) plus the Root Euclidean Distance (i.e., $d_{rootED}$), as this metric allows to compute the DELTACON similarity (i,e., $sim_{DC}$), which can quantify even small differences between two graphs in the interval [0, 1]. The pruning process involved removing a fraction of up to 10% of edges and nodes. This percentage has been chosen as the networks size was quite small (less than 250 nodes per each dataset).

Our analysis suggests that (i) the spectral metric $d_{\mathcal{L}}$ is best at catching the expected linear growth of differences with the incomplete graph against its complete counterpart; (ii) the node removal process is significantly more damaging than random edge removal; thus, it translates to a negligible error in terms of graph analysis when, for example, some wiretaps are missing.

Indeed, in terms of $sim_{DC}$ drop, there is a 30% difference from the real network, for a pruned version at 10%. On the other hand, it is crucial to be able to investigate the suspects in a timely fashion, since any exclusion of suspects from an investigation may lead to significant errors (due to substantial differences from the actual network)—we observed drops of up to 80% of $sim_{DC}$ on some networks.

A final consideration concerns the impossibility of conducting this type of analysis through the use of Machine Learning, as it is currently practically impossible to obtain a sufficient number of reliable and complete datasets of real criminal networks as to be able to conduct an appropriate training of a Neural Network.

For the future, we plan to extend the analysis by considering weights as well. This will allow to conduct a comparative analysis of the missing data effects when not only the connections between nodes, but also their frequency is known. Another interesting aspect to be considered is the network behaviour after their pruning in both criminal and general social networks. Lastly, using the future knowledge gained from the network analysis herein presented, one could try to define an artificial network able to accurately simulate the behavior of real criminal networks.

## Author Contributions

**Conceptualization:** Annamaria Ficara, Lucia Cavallaro, Francesco Curreri, Giacomo Fiumara, Pasquale De Meo, Antonio Liotta.

**Data curation:** Annamaria Ficara.

**Formal analysis:** Annamaria Ficara, Lucia Cavallaro, Francesco Curreri, Giacomo Fiumara, Ovidiu Bagdasar.

**Funding acquisition:** Wei Song, Antonio Liotta.

**Investigation:** Annamaria Ficara, Lucia Cavallaro, Giacomo Fiumara.

**Methodology:** Giacomo Fiumara, Pasquale De Meo.

**Project administration:** Lucia Cavallaro.

**Resources:** Annamaria Ficara.

**Software:** Annamaria Ficara, Lucia Cavallaro, Giacomo Fiumara.

**Supervision:** Pasquale De Meo, Ovidiu Bagdasar, Wei Song, Antonio Liotta.

**Validation:** Annamaria Ficara, Giacomo Fiumara.

**Visualization:** Annamaria Ficara, Lucia Cavallaro.

**Writing – original draft:** Annamaria Ficara, Lucia Cavallaro, Francesco Curreri.

**Writing – review & editing:** Pasquale De Meo, Ovidiu Bagdasar, Wei Song, Antonio Liotta.

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
