## [Decision Letter · Decision Letter 0]

5 May 2021

PONE-D-21-06688

Criminal networks analysis in missing data scenarios through graph distances

PLOS ONE

Dear Dr. Ficara,

Thank you for submitting your manuscript to PLOS ONE. After careful consideration, we feel that it has merit but does not fully meet PLOS ONE’s publication criteria as it currently stands. Therefore, we invite you to submit a revised version of the manuscript that addresses the points raised during the review process.

We look forward to receiving your revised manuscript.

Kind regards,

Hocine Cherifi

Academic Editor

PLOS ONE

Journal Requirements:

 [ ].

[ ].

Additional Editor Comments:

Reviewer #1: In the present manuscript, Ficara et al. apply several graph distance metrics to investigate the effect of missing nodes and edges on the interpretation of social networks of criminals; lack of nodes or edges is characteristic for these networks due to the nature of the data. The authors show that the distance metric best catching the expected differences in network structure is spectral distance based on the normalized Laplacian matrix. Using this metric, the authors show that the criminal networks are relatively robust against random removal of edges but highly vulnerable to random removal of nodes.

The manuscript is well-balanced and easy to read. The data used in described in detail, and the description of the network distance analysis is particularly clear. I am happy to recommend the manuscript for publication after the authors have addressed the few minor issues listed below.

Best regards,

Onerva Korhonen

post-doctoral researcher, Aalto University (Finland) / Universidad Politécnica de Madrid (Spain)

Minor comments:

1) Why do PK and SN differ so notably from the other networks in terms of degree distribution? Of course, it may be impossible to know for sure, but I would suggest using a couple of sentences for discussing the possible reasons.

2) What was the fraction of removed edges and nodes? The section "Design of experiments" (page 10) states that the networks were pruned according to a prefixed fraction (10%); however, Figs. 2, 3, and 4 suggest that instead of a fixed fraction, a range of fractions from 0 to 10% was used. In my opinion, using a fraction gives a better picture of the phenomenom, but the text should be modified accordingly.

3) I appreciate the public availability of the data used in the study. The algorithm used is relatively easy to reproduce based on the pseudocode provided; however, I still wonder if the actual implementation of the algorithm could be shared publicly as well.

Language-related issues:

1) In the abstract, "incorrectness, caused by either -- and --" should be "incorrectness, caused by either -- or --" or "incorrectness, caused by both -- and --".

2) In the abstract, "Our investigation identified" should probably be "Our investigation identifies" to avoid mixing past and present tense.

3) On page 2, line 10, SNA is given as an abbreviation for Network Science Analysis. It would feel more natural to use it for Social Network Analysis (or to use some other abbreviation for Network Science Analysis).

4) On page 12, the sentence "In order to quantify the difference between the original network and its pruned version, we computed several distance metrics, to the one which is most sensitive." is hard to follow, and the authors may wish to reformulate it.

Reviewer #2: I have studied the manuscript "Criminal networks analysis in missing data scenarios through graph distances" by Ficara and coauthors under consideration in PLOS ONE.

Here the authors present an analysis quantifying the effects of missing data on the study of criminal networks. To do so, the authors compare criminal networks with their pruned versions obtained after performing node or edge removal. By quantifying the difference between these two network versions, the analysis suggests that random node removal is more damaging than random edge removal. Thus, the authors conclude that incomplete data related to suspects significantly affects the study of criminal networks.

I have the following suggestion I wish the authors address before publication.

Although the distance metrics provide an approach to measure the effects of missing data on criminal networks, it is well-known that, compared to edge removal, node removal has a more significant impact on networks because the removal of a vertex results in the deletion of all edges attached to it. Hence, it is already expected that node removal will produce a greater difference between the original network and its pruned version.

To improve the robustness of their findings, I also suggest the authors compare these results with randomly generated graphs or another appropriate null model. This analysis might offer a more in-depth understanding of criminal networks by clarifying whether these networks are indeed more sensitive to node removal.

Typos:

- "Network Science Analysis (SNA)".

- "Read the dataset and covert it as graph G".
---

## [Author Response · Author response to Decision Letter 0]

24 Jun 2021

Reviewer 1

In the present manuscript, Ficara et al. apply several graph distance metrics to investigate the effect of missing nodes and edges on the interpretation of social networks of criminals; lack of nodes or edges is characteristic for these networks due to the nature of the data. The authors show that the distance metric best catching the expected differences in network structure is spectral distance based on the normalized Laplacian matrix. Using this metric, the authors show that the criminal networks are relatively robust against random removal of edges but highly vulnerable to random removal of nodes.

The manuscript is well-balanced and easy to read. The data used in described in detail, and the description of the network distance analysis is particularly clear. I am happy to recommend the manuscript for publication after the authors have addressed the few minor issues listed below.

Q1 Minor comments:

(a) Why do PK and SN differ so notably from the other networks in terms of degree distribution? Of course, it may be impossible to know for sure, but I would suggest using a couple of sentences for discussing the possible reasons.

(b) What was the fraction of removed edges and nodes? The section ''Design of experiments'' (page 10) states that the networks were pruned according to a prefixed fraction (10%); however, Figs. 2, 3, and 4 suggest that instead of a fixed fraction, a range of fractions from 0 to 10% was used. In my opinion, using a fraction gives a better picture of the phenomenon, but the text should be modified accordingly.

(c) I appreciate the public availability of the data used in the study. The algorithm used is relatively easy to reproduce based on the pseudocode provided; however, I still wonder if the actual implementation of the algorithm could be shared publicly as well.

A1 Our answers:

(a) We thank the Reviewer for their comment. We have discussed in a couple of sentences at the end of the Subsection ''Criminal networks data sources'' the possible reasons why PK and SN differ from the other networks.

(b) We thank the Reviewer for pointing out this sentence. We have now clarified it in the revised manuscript accordingly. Indeed, we removed one node per time up to the final pruning by reaching the 10% of nodes removed (or isolated) in total, at the end of the simulations.

(c) We thank the Reviewer for their note. They highlighted a precious aspect for the experiments repeatability. Indeed, we have now uploaded online on GitHub the source code of our algorithm and we have added the link (https://github.com/lcucav/criminal-nets/tree/master/missing_data) in the revised manuscript.

Q2 Language-related issues:

(a) In the abstract, ''incorrectness, caused by either -- and —'' should be ''incorrectness, caused by either -- or —'' or ''incorrectness, caused by both -- and —''.

(b) In the abstract, ''Our investigation identified'' should probably be ''Our investigation identifies'' to avoid mixing past and present tense.

(c) On page 2, line 10, SNA is given as an abbreviation for Network Science Analysis. It would feel more natural to use it for Social Network Analysis (or to use some other abbreviation for Network Science Analysis).

(d) On page 12, the sentence ''In order to quantify the difference between the original network and its pruned version, we computed several distance metrics, to the one which is most sensitive.'' is hard to follow, and the authors may wish to reformulate it.

A2 We thank the Reviewer for their valuable comments. We have corrected all the language-related issues.

—————————————————————

Reviewer 2

I have studied the manuscript ''Criminal networks analysis in missing data scenarios through graph distances'' by Ficara and coauthors under consideration in PLOS ONE.

Here the authors present an analysis quantifying the effects of missing data on the study of criminal networks. To do so, the authors compare criminal networks with their pruned versions obtained after performing node or edge removal. By quantifying the difference between these two network versions, the analysis suggests that random node removal is more damaging than random edge removal. Thus, the authors conclude that incomplete data related to suspects significantly affects the study of criminal networks.

I have the following suggestion I wish the authors address before publication.

Q1 Although the distance metrics provide an approach to measure the effects of missing data on criminal networks, it is well-known that, compared to edge removal, node removal has a more significant impact on networks because the removal of a vertex results in the deletion of all edges attached to it. Hence, it is already expected that node removal will produce a greater difference between the original network and its pruned version.

A1 We thank the Reviewer for their note. Being it extremely on point, we have mentioned it as well in the Subsection ''Design of the Experiments'', following their suggestion.

Q2 To improve the robustness of their findings, I also suggest the authors compare these results with randomly generated graphs or another appropriate null model. This analysis might offer a more in-depth understanding of criminal networks by clarifying whether these networks are indeed more sensitive to node removal.

A2 We thank the Reviewer for such note that shows genuine interest to our work. In our previous work [1] we indeed applied graph distances to compare artificial and criminal networks in the attempt to explain how artificial models may be able to mirror the topology of real criminal networks. Experiments were performed showing how the distance varies as the structure of different artificial Barabasi-Albert models changes to match the one of the real criminal network used as comparison. We added a note in the revised manuscript as well to highlight these previous experiments, as kindly suggested.

Q3 Typos:

''Network Science Analysis (SNA)''.

''Read the dataset and covert it as graph G''.

A3 We thank the Reviewer for their valuable comments. We have corrected all the typos.

[1] Lucia Cavallaro, Annamaria Ficara, Francesco Curreri, Giacomo Fiumara, Pasquale De Meo, Ovidiu Bagdasar, and Antonio Liotta. Graph comparison and artificial models for simulating real criminal networks. In Rosa M. Benito, Chantal Cherifi, Hocine Cherifi, Esteban Moro, Luis Mateus Rocha, and Marta Sales-Pardo, editors,Complex Networks andTheir Applications IX, pages 286–297, Cham, 2021. Springer International Publishing.

---

## [Decision Letter · Decision Letter 1]

9 Jul 2021

Criminal networks analysis in missing data scenarios through graph distances

PONE-D-21-06688R1

Dear Dr. Liotta,

We’re pleased to inform you that your manuscript has been judged scientifically suitable for publication and will be formally accepted for publication once it meets all outstanding technical requirements.

Kind regards,

Hocine Cherifi

Academic Editor

PLOS ONE

Reviewers' comments:

Reviewer's Responses to Questions

**Comments to the Author**

1. If the authors have adequately addressed your comments raised in a previous round of review and you feel that this manuscript is now acceptable for publication, you may indicate that here to bypass the “Comments to the Author” section, enter your conflict of interest statement in the “Confidential to Editor” section, and submit your "Accept" recommendation.

Reviewer #2: All comments have been addressed

2. Is the manuscript technically sound, and do the data support the conclusions?

Reviewer #2: Yes

3. Has the statistical analysis been performed appropriately and rigorously? 

Reviewer #2: Yes

4. Have the authors made all data underlying the findings in their manuscript fully available?

Reviewer #2: Yes

5. Is the manuscript presented in an intelligible fashion and written in standard English?

Reviewer #2: Yes

6. Review Comments to the Author

Reviewer #2: (No Response)

7. PLOS authors have the option to publish the peer review history of their article (what does this mean?). If published, this will include your full peer review and any attached files.

Reviewer #2: **Yes: **Alvaro Franco Martins

---

## [Editor Report · Acceptance letter]

15 Jul 2021

PONE-D-21-06688R1 

Criminal networks analysis in missing data scenarios through graph distances 

Dear Dr. Liotta:

I'm pleased to inform you that your manuscript has been deemed suitable for publication in PLOS ONE. Congratulations! Your manuscript is now with our production department. 

Kind regards, 

on behalf of

Professor Hocine Cherifi 

Academic Editor

PLOS ONE